# Fluorination-enabled optimal morphology leads to over 11% efficiency for inverted small-molecule organic solar cells

Dan Deng[1,*], Yajie Zhang[1,*], Jianqi Zhang[1], Zaiyu Wang[2], Lingyun Zhu[1], Jin Fang[1,3], Benzheng Xia[1], Zhen Wang[1], Kun Lu[1], Wei Ma[2] & Zhixiang Wei[1,3]

Solution-processable small molecules for organic solar cells have attracted intense attention for their advantages of definite molecular structures compared with their polymer counterparts. However, the device efficiencies based on small molecules are still lower than those of polymers, especially for inverted devices, the highest efficiency of which is <9%. Here we report three novel solution-processable small molecules, which contain π-bridges with gradient-decreased electron density and end acceptors substituted with various fluorine atoms (0F, 1F and 2F, respectively). Fluorination leads to an optimal active layer morphology, including an enhanced domain purity, the formation of hierarchical domain size and a directional vertical phase gradation. The optimal morphology balances charge separation and transfer, and facilitates charge collection. As a consequence, fluorinated molecules exhibit excellent inverted device performance, and an average power conversion efficiency of 11.08% is achieved for a two-fluorine atom substituted molecule.

[1] CAS Key Laboratory of Nanosystem and Hierarchical Fabrication, CAS Center for Excellence in Nanoscience, National Center for Nanoscience and Technology, Beijing 100190, China. [2] State Key Laboratory for Mechanical Behavior of Materials, Xi'an Jiaotong University, Xi'an 710049, China. [3] University of Chinese Academy of Sciences, Beijing 100049, China. * These authors contributed equally to this work. Correspondence and requests for materials should be addressed to K.L. (email: lvk@nanoctr.cn) or to W.M. (email: msewma@xjtu.edu.cn) or to Z.W. (email: weizx@nanoctr.cn).

Organic solar cells (OSCs) have attracted intense attentions due to their potential for solution processing cheap and flexible devices. Comparing with conventional device architectures, inverted devices exhibit improved environmental stability and more preferable for industrial applications[1]. Polymer solar cells with power conversion efficiency (PCE) higher than 10% are mainly obtained through inverted devices[2–4], and the highest efficiency has reached 11.7% (ref. 5). Solution-processable small molecules for OSCs have attracted intense attentions for their advantages of high purity and definite molecular structures compared with polymers[6–11]. To date, the PCE based on small molecules has reached 10% by conventional devices[8,12]. However, attempts to develop inverted devices for small molecules are not as successful as that of polymers[13–16], and the highest PCE reported is 8.84% (ref. 17), which is much lagging behind their polymer couterparts.

To obtain highly efficient OSCs, decisive parameters, namely, open circuit voltage ($V_{oc}$), fill factor (FF) and short-circuit current ($J_{sc}$) should be enhanced. Each parameter could be expressed as following general formula:

$$\Lambda = \Lambda_{max} - \Lambda_{loss}(\Lambda = V_{oc}, \text{FF or } J_{sc}) \qquad (1)$$

A proved successful strategy in molecular design is to increase $\Lambda_{max}$. The approaches involve increasing ionization potential of the donor ($IP_D$), narrowing bandgap and enhancing mobility to achieve high $V_{oc}$, $J_{sc}$ and FF, respectively[18–20]. However, $IP_D$ enhancement would lead to bandgap increase[19], and $(V_{oc} \times J_{sc})_{max}$ is limited by Shockley–Queisser model. To further maximize PCE, an alternative strategy is to minimize $\Lambda_{loss}$. The main loss of $V_{oc}$ is related to the disorder arrangement of donors/acceptors and their poor contact with electrode[21–23]; the main losses of FF and $J_{sc}$ are ascribed to the recombination induced by undesirable distribution of intermix and crystalline phases, interfacial traps and poor domain purity[24,25]. Consequently, all losses are related to the bulk-heterojunction (BHJ) morphology, the optimization of which could simultaneously minimize $\Lambda_{loss}$ for all three parameters.

The adjustment of the donor–acceptor (D:A) interaction is an essential approach to optimize the BHJ morphologies. A difference in surface free energies between donors and acceptors would result in a repulsive interaction (lower miscibility), which acts as the internal driving force to form phase separation[26,27]. Hence, a proper disparity of surface free energies is significant to achieving optimized lateral morphology. Furthermore, a lower surface free energy of donors in comparison with acceptors would induce surface enrichments and vertical phase separation[28], both of which could effectively decrease losses of $V_{oc}$, $J_{sc}$ and FF through suppression of recombination by modifying interface contact, forming charge-blocking regions and facilitating charge collection[24,28].

In this paper, three medium bandgap molecules are designed and synthesized with thiophene-substituted benzodithiophene (TBDT) as a core, 2-(thiophen-2-yl)thieno [3,2-b]thiophene as π-bridges and end-capped with 1H-indene-1,3(2H)-dione, 4-fluoro-1H-indene-1,3(2H)-dione or 4,7-difluoro-1H-indene-1,3(2H)-dione; these molecules are abbreviated as **BTID-0F**, **BTID-1F** and **BTID-2F** (Fig. 1a). With incremental introduction of fluorine to end-capped units, the PCE for inverted devices increases from 8.30% for **BTID-0F** to 10.4% for **BTID-1F** and to 11.3% for **BTID-2F**. A hierarchical morphology with higher domain purity, enhanced surface enrichment and more directional vertical phase distribution is induced by fluorination, thereby $V_{oc}$, $J_{sc}$ and FF are increased simultaneously.

## Results

**Design and characterization of small molecules.** To benefit the charge carrier transport, a two-dimensional donor unit, TBDT was selected as the core[29]. A new π-bridge, namely, 2-(thiophen-2-yl)thieno[3,2-b]thiophene, was designed with two advantages: first, it introduces stronger aromaticity units of thieno [3,2-b]thiophene to increase $IP_D$, aiming to increase attainable $V_{oc}$. Second, it presents an inner gradient-decreased electron density distribution (Supplementary Fig. 1), benefited for backbone hole transfer. To lower the miscibility and increase dielectric constant, 1H-indene-1,3(2H)-dione was selected as acceptor because of its stronger polarity and aromatic difference in comparison with [6,6]-phenyl-$C_{71}$-butyric acid methyl ester ($PC_{71}BM$)[26]. To decrease molecular surface free energies and miscibility with $PC_{71}BM$, we introduced fluorine atoms[30] to the end-capped acceptor units and obtained two new acceptor units, namely, 4-fluoro-1H-indene-1,3(2H)-dione and 4,7-difluoro-1H-indene-1,3(2H)-dione. Various studies have discussed fluorination effects on photovoltaic performance but most focused on modifying molecular internal part to stabilize molecular conformation[31,32]. In addition, the positions of alkyl chains were designed to prevent backbone torsion to highest extent. The different alkyl chains attached to TBDT units for **BTID-0F**, **BTID-1F** and **BTID-2F** were used to ensure their sufficient solubility. The synthesis routes of the three molecules are shown in Supplementary Fig. 2. The dielectric constants of **BTID-0F**, **BTID-1F** and **BTID-2F** are calculated as ca. 4.0 (Supplementary Table 1, Supplementary Fig. 3a), which was a high value among the conjugated small molecules or polymers[33].

Absorption spectra of solution and films are shown in Fig. 1b. The absorption coefficient in solution of the three molecules denoted the competition between the content of alkyl chains and fluorine atoms. Although, four fluorine atoms were introduced into **BTID-2F**, the alkyl chain (no contribution to absorption) content of which was the highest. Thus, **BTID-2F** exhibited the lowest absorption coefficient in solution, whereas **BTID-1F** showed the highest. Furthermore, introduction of fluorine not only enhanced absorption coefficient gradually but also slightly redshifted the absorption spectrum. From solution to films, all the molecules clearly manifested ca. 70 nm redshifts, illustrating good aggregation in films. With the increase of fluorine atoms, the intensity ratios of the shoulder peak (attributed to π–π stacking) to the absorption peak from internal charge transfer increased. The different tendencies of film absorption coefficient and the solution absorption coefficient further revealed that fluorination led to dense and ordered π–π stacking.

Ultraviolet photoelectron spectroscopy (UPS) was carried out to obtain $IP_D$ in pristine films and blends with $PC_{71}BM$ on PEDOT:PSS/ITO substrates (Fig. 1c)[34], and the IP and EA (electron affinity) of $PC_{71}BM$ were obtained from literatures[35]. The $IP_D$ of the three pristine films were 4.91, 4.98 and 5.05 eV, respectively, indicating fluorination of the end acceptors increase $IP_D$ comparing with those of fluorine atoms attached to the internal part of polymers[36,37]. The detailed $-IP_D$ and $-EA_D$ of the three molecules are shown Fig. 1d, and the $EA_D$ was calculated from: $EA_D = -(E_g^{opt} - IP_D)$ ($E_g^{opt}$ is optical energy bandgap). However, after blending with $PC_{71}BM$, the $IP_D$ were changed to 4.98, 4.92 and 4.95 eV for **BTID-0F**, **BTID-1F** and **BTID-2F**, and their corresponding changed values were 0.09, $-0.05$ and $-0.09$ eV. The change from pristine to blended films was mainly due to the shifts in work function (Fig. 1c) indicating dipole polarity differs after $PC_{71}BM$ addition. Changes in dipole direction after fluorination were probably due to various surface enrichment and vertical distributions, which will be discussed in the later section. Moreover, the molecules based on the new designed π-bridge exhibited low highest occupied molecular

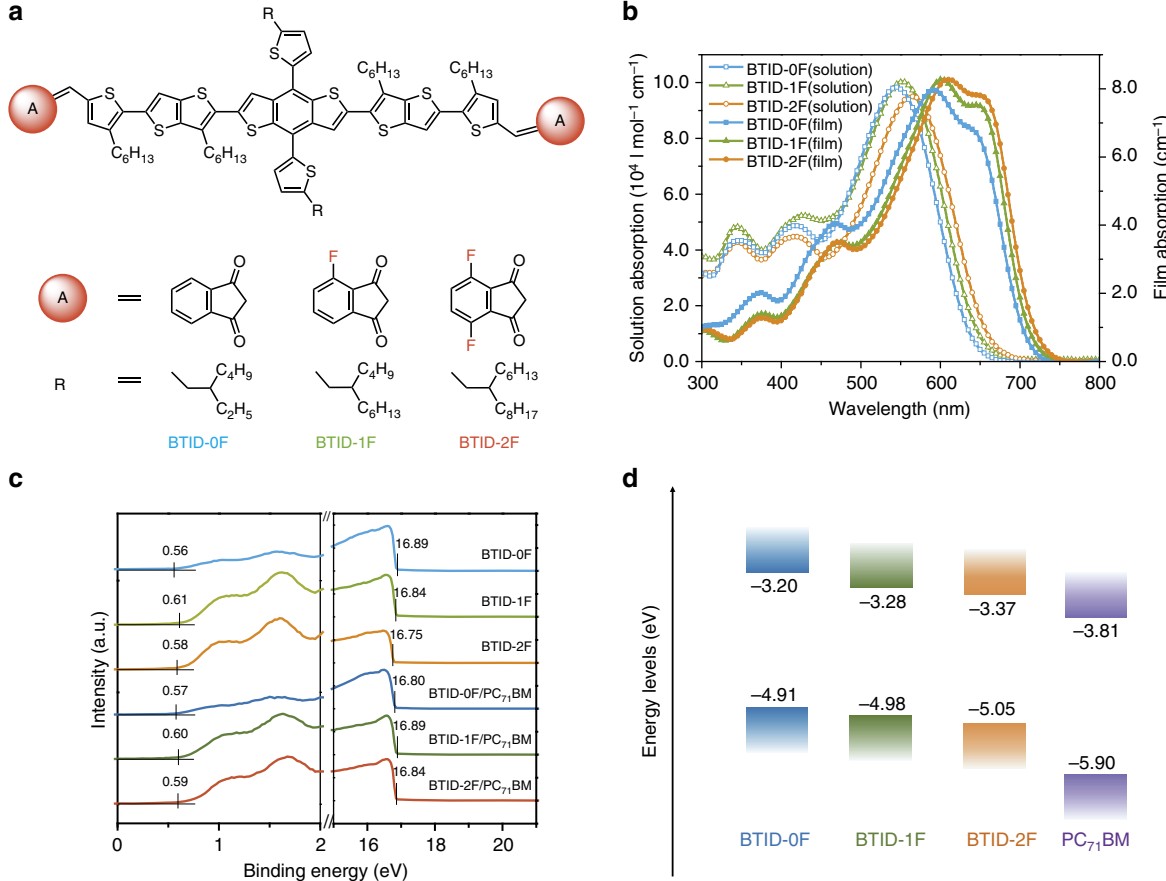

**Figure 1 | Molecular structures and properties.** (**a**) Small molecular structures; (**b**) solution coefficient of samples in chloroform (left Y axis) and film (right Y axis); (**c**) UPS results of the pristine films and blended films with $PC_{71}BM$; and (**d**) molecular energy levels measured from UPS, the lowest unoccupied molecular orbital (LUMO) levels calculated from optical bandgap and UPS.

orbital (HOMO) levels (measured by cyclic voltammetry, Supplementary Fig. 3b,c), which is the basis for obtaining high $V_{oc}$ in BHJ OSCs[5].

To check the miscibility and molecular surface free energies, the contact angles of chloroform solution of three materials were measured on ZnO/ITO and PEDOT/ITO substrates (Supplementary Fig. 3d,e). The increase of contact angles for the fluorinated molecules indicated fluorination lowered the surface free energies of small molecules. On the other hand, the increase of deviation in contact angles between fluorinated molecules and $PC_{71}BM$ indicated fluorination also reduced the miscibility of small molecules with $PC_{71}BM$. These results implied that surface free energies and miscibility were successfully adjusted through molecular design.

**Fabrication and performance of inverted solar cells**. To investigate the photovoltaic properties of the three small molecules, devices with a structure of ITO/ZnO/active layer/MoO$_x$/Ag were fabricated (Fig. 2a). The highest PCE for **BTID-0F**, **BTID-1F** and **BTID-2F** were 8.30, 10.4 and 11.3%, and other detailed parameters are shown in Fig. 2b,c and Table 1, and the optimization process of D:A ratios is shown in Supplementary Table 2. The device based on **BTID-2F** was certified at an accredited laboratory, certifying a PCE of 11.0% (Supplementary Fig. 4a–e). Notably, the active layers of all devices were obtained without any additives and post-treatment, which were facilitated for future industrial manufacturing[38].

The high efficiencies of the three materials were ascribed to simultaneous increment in $V_{oc}$, $J_{sc}$ and FF. The high $V_{oc}$ was in

good agreement with the $IP_D$ of their blends obtained by UPS. As calculated from the formula: $eV_{oc} = IP_D - EA_A - \Delta V$ (refs 39,40), the $V_{oc}$ losses ($\Delta V$) based on all three small molecules were *ca.* 0.2 eV ($-IP_D$ and $-EA_A$ values shown in Fig. 1d), which were much less than empirical value $0.3 - 0.5$ eV reported[39,40]. The low $V_{oc}$ loss were attributed to their high dielectric constant (Supplementary Fig. 3a, Supplementary Table 1)[33,40] and ideal morphologies, which will be discussed in following sections. The $J_{sc}$ of **BTID-2F** and **BTID-1F** cells was higher than that of **BTID-0F** cells partly because of the larger photocurrent generated in the red region (Fig. 2c). On the other hand, the enhancements of FF and partial $J_{sc}$ for **BTID-2F** and **BTID-1F** compared with **BTID-0F**, illustrated lower loss of FF and $J_{sc}$ with fluorination. This phenomenon could be supported by the relation of photocurrent density ($J_{ph}$) versus effective voltage ($V_{eff}$) (Fig. 2d), where $J_{ph} = J_L - J_D$ ($J_L$ and $J_D$ are the current density under illumination and in the dark) and $V_{eff} = V_0 - V_a$, ($V_0$ is the voltage at $J_{ph} = 0$ and $V_a$ is the measured voltage under different current density). The ratios of $J_{ph}/J_{ph,sat}$ are used to judge the overall efficiency of exciton dissociation and charge collection[2,41]. Under short-circuit condition, the ratios were 0.93, 0.95 and 0.96 for **BTID-0F**, **BTID-1F** and **BTID-2F**, suggesting the effective exciton dissociation of the three molecules, especially for **BTID-1F** and **BTID-2F**. Under maximal power output circumstances ($V_{eff} = 0.2$ V), the ratios of $J_{ph}/J_{ph,sat}$ were 0.71, 0.83 and 0.86, respectively, indicating a considerably higher charge collection and lower bimolecular recombination after fluorine substitution. The superior $J_{ph}-V_{eff}$ characteristics clearly demonstrated that

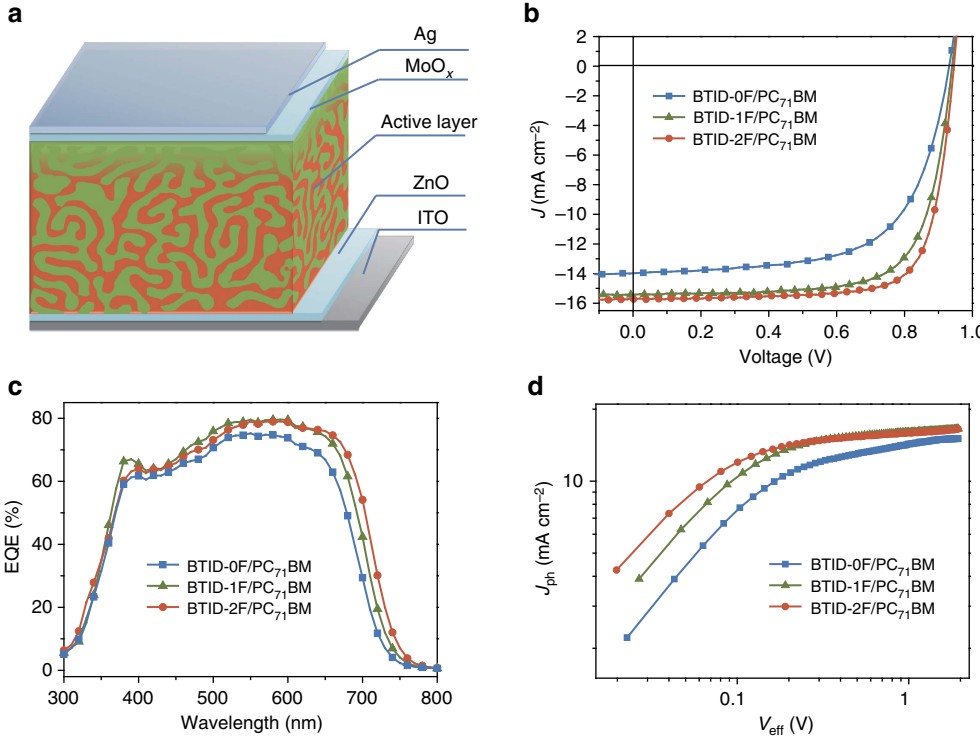

**Figure 2 | Device structures and photovoltaic properties for inverted solar cells.** (**a**) Structures of inverted device; (**b**) optimized J–V curves for inverted devices; (**c**) EQE corresponding to devices in **b**,**d** photocurrent density versus effective voltage ($J_{ph}$–$V_{eff}$) characteristics for devices under constant incident light intensity (AM 1.5 G, 100 mW cm$^{-2}$).

**Table 1 | Optimized photovoltaic performance of inverted device based on the three molecules (device structure: ITO/ ZnO/active layer/MoO$_x$/Ag).**

| Donors | $V_{oc}$ (V) | $J_{sc}$ (mA cm$^{-2}$) | FF (%) | PCE (%) | |
|---|---|---|---|---|---|
| | | | | Best | Average |
| BTID-0F | 0.93 | 14.0 | 64.0 | 8.30 | 8.21 |
| BTID-1F | 0.94 | 15.3 | 72.0 | 10.4 | 10.37 |
| BTID-2F | 0.95 | 15.7 | 76.0 | 11.3 | 11.08 |

fluorination could reduce bimolecular recombination, thereby improving $J_{sc}$ and FF simultaneously.

**Hierarchical morphology**. The small loss in $J_{sc}$, FF and $V_{oc}$ for devices based on the three molecules and their variation in loss should be ascribed to the optimized but different lateral/vertical phase distributions. To study morphologies in the lateral direction, we characterized small molecules/PC$_{71}$BM blending films on ZnO/Si substrates and pristine films on Si substrates by grazing incidence wide-angle X-ray scattering (GIWAXS) (Fig. 3a–f), and their corresponding one-dimension curves are shown in Fig. 3g,h. Whether in pristine or blended films, all three molecules exhibited preferable edge-on molecular packing orientation with a small ratio of face-on packing orientation, because evident multiple higher-order ($h$00) reflections in the out-of-plane direction and an evident (010) reflection of π–π stacking in the in-plane direction were observed for all samples. From the calculated face-on to edge-on ratios (Supplementary Table 3), it could be easily found: in pristine films, molecules adopted a more favourable edge-on packing mode; while in blends, the ratios of face-on to edge-on orientation were similar for the three molecules.

The differences in $d$-spacing in the (100) direction were ascribed to the varying lengths of the alkyl (corresponds to the short axis periodicity). The π–π stacking distance of **BTID-0F**, **BTID-1F** and **BTID-2F** were 3.63, 3.57 and 3.55 Å, illustrating a more condensed stacking in the π–π direction after fluorination; a result was consistent with the absorption in films. The coherence lengths calculated from the π–π stacking (010) peaks were 51.0, 67.3 and 70.4 Å for **BTID-0F**, **BTID-1F** and **BTID-2F**, suggesting fluorination of end-capped units increases order range. In comparison with pristine film (Fig. 3a–c), additions of PC$_{71}$BM decreased the coherence length by 13.2, 10.4 and 2% for **BTID-0F**, **BTID-1F** and **BTID-2F**, indicating that fluorination decreases the influence of PC$_{71}$BM on molecular aggregation, especially for **BTID-2F**; the phenomenon could be ascribed to their decreased miscibility with PC$_{71}$BM. In addition, PC$_{71}$BM in all blends exhibited strong aggregation with coherence lengths ca. 20 Å (Supplementary Table 3). The good aggregations for PC$_{71}$BM and small molecules reduced interfacial energy disorder, which was beneficial for further lowering loss of $V_{oc}$ (ref. 22).

The morphologies were further investigated by atomic force spectroscopy (AFM) and transmission electron microscopy (TEM). The domain in AFM phase images (Fig. 4a–c) increased in size after fluorinations in accordance with their enhanced crystalline. As seen in TEM images (Fig. 4d–f), nanostructures were observed for **BTID-0F**/PC$_{71}$BM blends with diameters ca. 25 nm. After fluorination, the diameters of nanostructure increased evidently for **BTID-1F**/PC$_{71}$BM blends because of increased order range. Interestingly, with a further increase in fluorine atoms, an evident network of whiskers with diameter ca. 15 nm was observed, interpenetrating in the larger domains.

Resonant soft X-ray scattering (RSoXS) was employed to investigate the above mentioned fine microstructure (Fig. 4g). With increment of fluorine atoms, the dominated domain size increased from 24 to 43, and finally to 53 nm, as in good

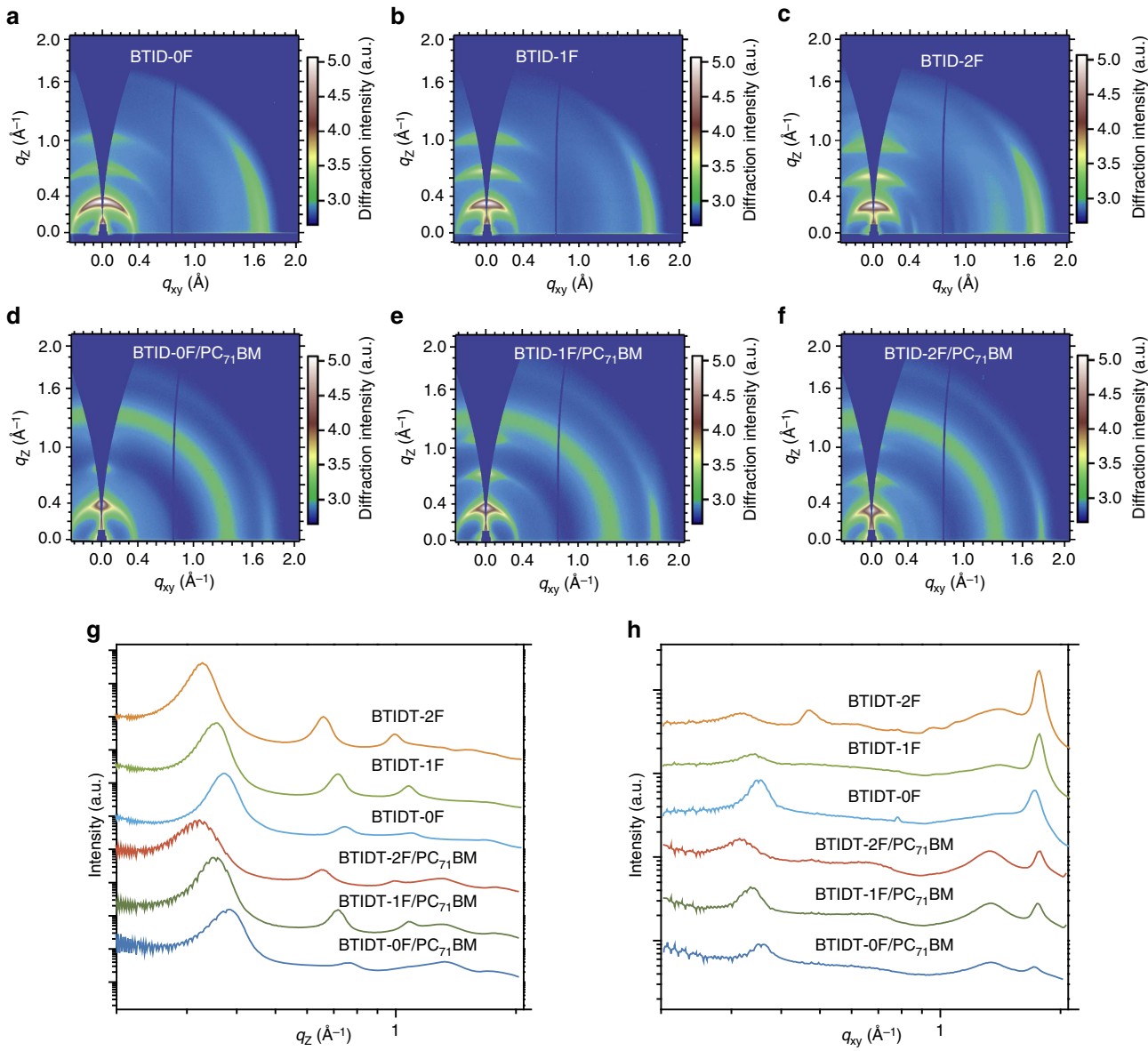

**Figure 3 | Microstructures of pristine and blend films.** (**a**–**f**) GIWAXS images in pristine films on Si substrate and GIWAXS images in blends films on ZnO/Si substrates; (**g**) corresponding out-of-plane curves; and (**h**) corresponding in-plane curves.

agreement with the increased crystallinity. However, by carefully analysing the RSoXS profiles, we could see that the scattering distribution could be fitted by two log-normal functions, with the other peak in the longer $q$-values: $ca.$ $0.25\,nm^{-1}$ for **BTID-2F** (corresponding to domain size of 12.8 nm, calculated from plot fitting in Supplementary Fig. 5) and a less evident interference appeared at $ca.$ $0.22\,nm^{-1}$ for **BTID-1F** (corresponding to domain size of 14.4 nm). As for **BTID-0F**, no obvious difference was found between the two fitting domain sizes (24 and 20 nm). In other words, **BTID-2F**/PC$_{71}$BM and **BTID-1F**/PC$_{71}$BM blends demonstrated formation of a hierarchical morphology with secondary domain sizes, and the secondary domain size was closed to the exciton diffusion length of $ca.$ 10 nm (Fig. 4h). Moreover, the relative domain purity for **BTID-0F**, **BTID-1F** and **BTID-2F** was calculated as 0.70, 0.93 and 1, in good agreement with increased crystallinity and decreased miscibility by increased fluorine introduction (Fig. 4g,h). The hierarchical morphology is reported to well balance domain size and purity to facilitate charge separation and transfer; the smaller donor phase accounts

for charge separation, whereas the larger donor phase is responsible for charge transport[42,43]. Therefore, the hierarchical morphology consisting 10–20 nm structure and enhanced domain purity could increase FF and $J_{sc}$ simultaneously.

**Surface enrichment and vertical phase distribution**. X-ray photoelectron spectroscopy (XPS) was carried out to study surface enrichment and vertical phase distribution. In the blends of small molecules: PC$_{71}$BM, sulfur and fluorine atoms could be used as characteristic elements of the small molecules due to the absence of the two elements of PC$_{71}$BM. A parameter named 'surface enrichment degree' was introduced to characterize surface enrichment, which was equal to divide the S:C (or F:C) ratio measured through XPS by the ideal S:C (or F:C) ratio calculated from D:A ratio (Fig. 5a). The histogram in Fig. 5a illustrated that all three molecules were prone to accumulating on the surface. With fluorination, surface enrichment was enhanced from 1.69 for **BTID-0F** to 1.79 for **BTID-1F** and 1.81 for **BTID-2F**, as

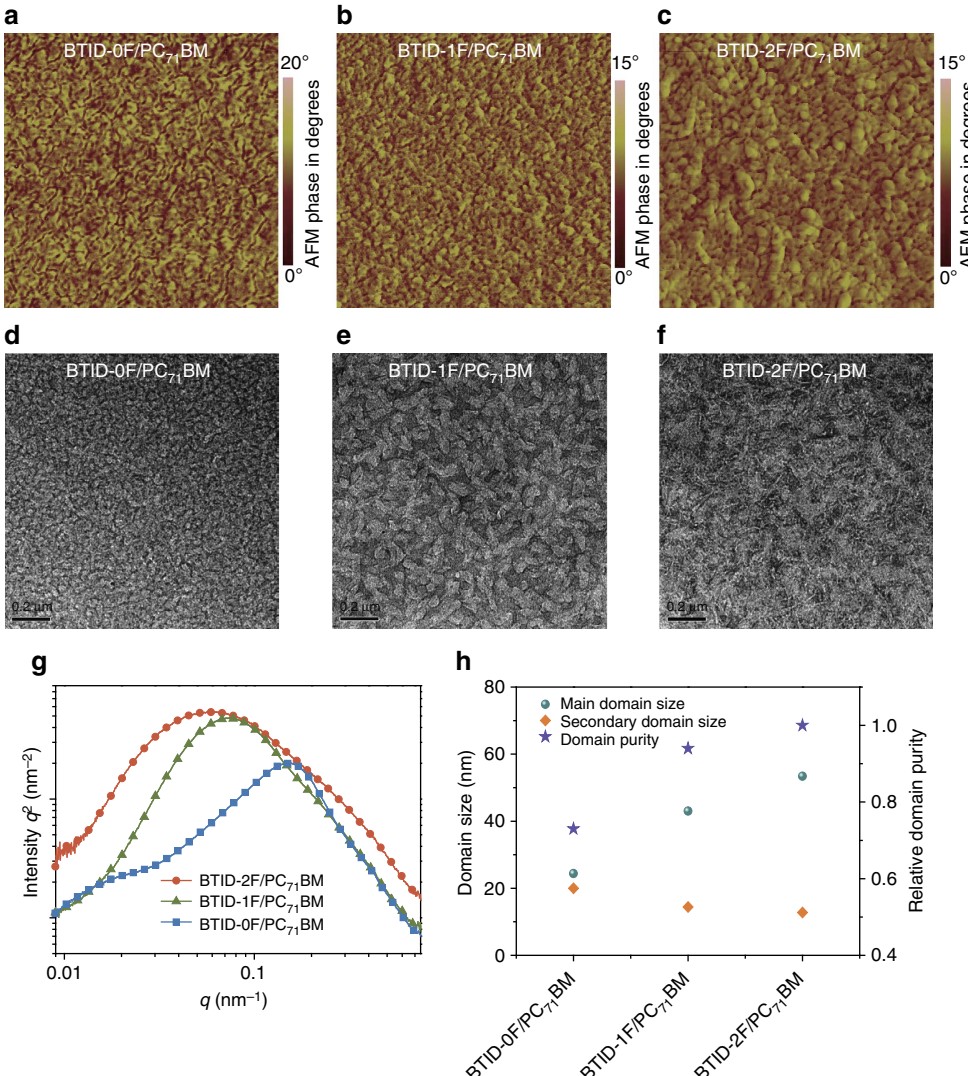

**Figure 4 | Lateral morphologies and microstructures of blend films on ZnO/ITO substrates.** (**a–c**) AFM phase images; (**d–f**) TEM images; (**g**) RSoXs profiles; and (**h**) relative domain purity and domain sizes.

calculated from S:C ratios. The 'surface enrichment degree' calculated from F:C ratios were markedly higher (2.1 for **BTID-1F** and 2.6 for **BTID-2F**) than those of S:C ratios, indicating that the fluorine-substituted groups are more prone to be enriched at the active layer surface.

Subsequently, in-depth XPS measurements in the vertical direction of **BTID-0F** and **BTID-2F** blends on ZnO/ITO substrates were carried out to characterize the vertical phase distribution (Fig. 5b). We defined the active layer/ZnO interface by the appearance of a large amount of Zn element. The corrosion time of active layer was slightly different because of the material and film thickness differences. **BTID-0F** exhibited enrichments both on top surface (air/active layer) and bottom interface (active layer/ZnO). As for **BTID-2F**, surface enrichment degree was increased evidently, whereas enrichment at the bottom and bulk was suppressed. As calculated from F:C ratios, the top surface was nearly 100% of **BTID-2F** (Supplementary Methods), indicating an electron blocking layer was formed at the active layer/MoO$_x$ interface. The more directional vertical phase distribution and formation of electron blocking layer facilitated the charge extraction/collection and recombination suppression, leading to a higher FF and $J_{sc}$ in inverted devices than those of conventional devices for **BTID-2F** (Supplementary Fig. 6,

Supplementary Table 4). Moreover, the vertical distribution and surface enrichment reduced recombination and increased the build-in potential of inverted devices in comparison of its conventional devices, which thereby further reduced $V_{oc}$ loss. The influence of surface enrichment and vertical phase separation on the device performance was further certified in the Supplementary evidence of the role of surface enrichment and vertical phase separation (Supplementary Discussions, Supplementary Fig. 7, Supplementary Tables 3 and 5).

The surface enrichment and vertical phase distribution were further manifested by results of charge carrier mobility (Fig. 5c). All the carrier mobility were used average values measured by space-charge limited current method (Supplementary methods), obtained from thickness between 100 and 130 nm (Supplementary Fig. 8). The average hole mobility of pristine film for **BTID-0F**, **BTID-1F** and **BTID-2F** were $8.7 \times 10^{-4}$ cm$^2$ V$^{-1}$ s$^{-1}$, $6.4 \times 10^{-4}$ cm$^2$ V$^{-1}$ s$^{-1}$ and $3 \times 10^{-4}$ cm$^2$ V$^{-1}$ s$^{-1}$, the decrease of hole mobility with fluorination should be mainly resulted from their decreased face-on packing ratios (Supplementary Table. 3) and longer alkyl chain lengths. However, after blending of PC$_{71}$BM, **BTID-2F** showed markedly higher hole mobility than **BTID-0F**, the average of which were $1.4 \times 10^{-3}$ cm$^2$ V$^{-1}$ s$^{-1}$ and $4.7 \times 10^{-4}$ cm$^2$ V$^{-1}$ s$^{-1}$. On the

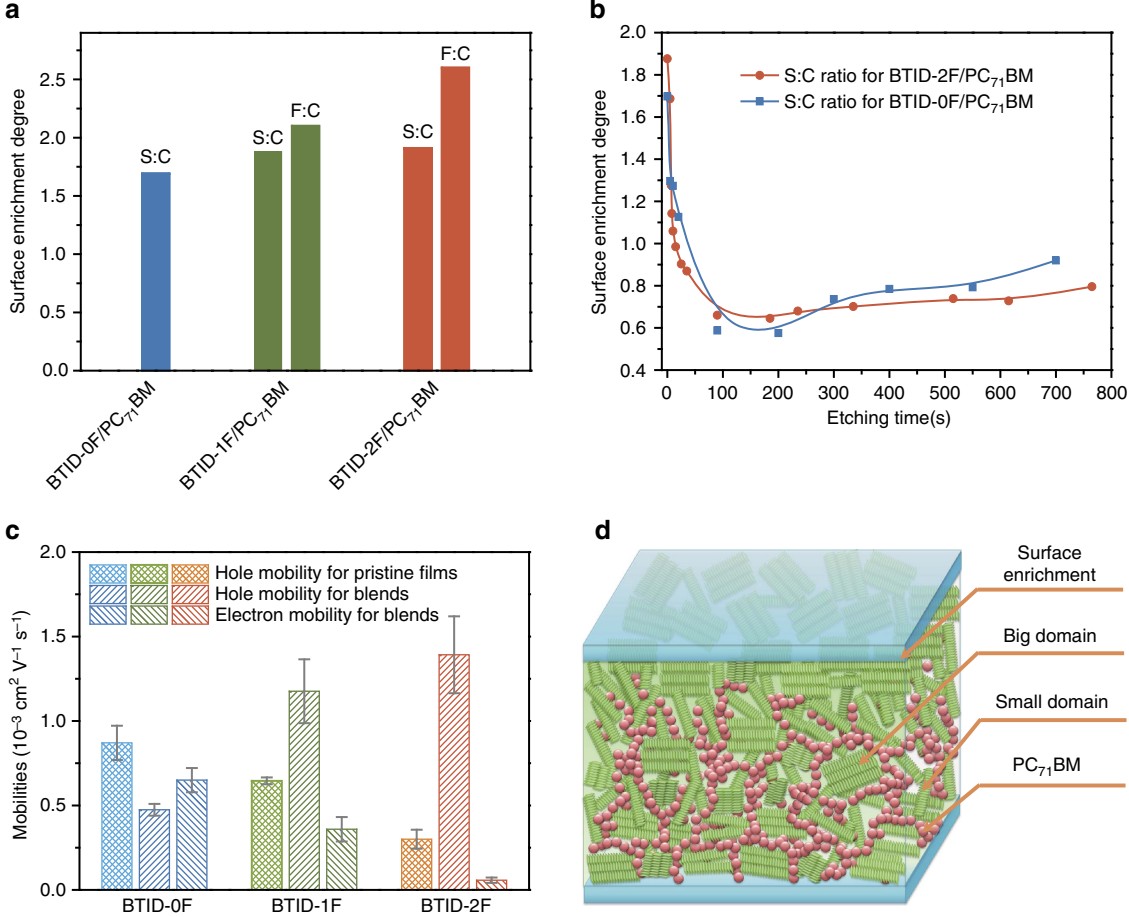

**Figure 5 | Vertical morphologies of blended films.** (**a**) Surface enrichments of BTID-0F/PC$_{71}$BM, BTID-1F/PC$_{71}$BM and BTID-2F/PC$_{71}$BM on ZnO/ITO substrate; (**b**) in-depth XPS profiles of BTID-0F/PC$_{71}$BM and BTID-2F/PC$_{71}$BM on ZnO/ITO substrates, the line were obtained from fitting all the S:C value from various etching time; (**c**) carrier mobilities of blends and pristine films, the average values were obtained from the mobility measured from the thickness between 100 and 130 nm, the error bars come from the mobility value from thickness differences and their measurements errors; and (**d**) schematic illustrations of lateral and vertical phase distribution of BTID-2F/PC$_{71}$BM.

other hand, the electron mobility of **BTID-2F**/PC$_{71}$BM blends was considerably lower than that of **BTID-0F**/PC$_{71}$BM blends. The hole and electron mobility of **BTID-1F**/PC$_{71}$BM were in between. The different trends for hole and electron mobility could be explained by the largest surface enrichment of **BTID-2F**, which facilitated hole transport while blocked electron transport. The increase in hole mobility of blends compared with pristine films for fluorine-substituted molecules could be explained by their increased ratios of face-on to edge-on after PC$_{71}$BM blending (Supplementary Table 3) and favourable backbone orientation for vertical charge transport[10].

## Discussion

Based on hierarchical morphology at lateral direction and vertical phase distribution, the morphology of active layer was schematically shown as Fig. 5d for **BTID-2F**/PC$_{71}$BM. In lateral direction, **BTID-2F** formed a hierarchical morphology with high domain purity, composing domains with diameters of *ca.* 53.0 nm and an interpenetrating whisker network with diameters of *ca.* 12.8 nm. A similar hierarchical morphology for **BTID-1F**/PC$_{71}$BM was observed but with a lower domain purity, while no obvious difference for two domain sizes in **BTID-0F**/PC$_{71}$BM. In vertical direction, **BTID-2F** formed hole-transporting layers on the top interface and more directional vertical phase distributions of the active layer than those of **BTID-0F**.

Therefore, fluorination-enabled optimal hierarchical morphology and surface enrichment, which could increase $V_{oc}$, $J_{sc}$ and FF simultaneously and thereby obtained a high PCE in inverted devices. To further verify the influence of hierarchical morphology on device performance (Supplementary Discussions), the substrate temperature was increased from 28 °C (normal) to 40 °C (hot) during film formation. Due to a faster solvent evaporation on hot substrate, the hierarchical morphology was not formed as proved by TEM images and RSoXS images (Supplementary Fig. 9). As a consequence, the device performance based on the hot substrate was decreased (Supplementary Table 6). Hence, the hierarchical morphology is one of the most important factors to obtain high performance devices.

For the molecular design, acceptor–donor–acceptor have been widely used for organic photovoltaic. To obtain a low HOMO level and a high hole mobility simultaneously, a novel π-bridge between donor and acceptor unit, 2-(thiophen-2-yl) thieno [3,2-*b*]thiophene was introduced, which presented an inner gradient-decreased electron density distribution, and facilitated the backbone charge transfer. Different from that of polymers, the end acceptor played an important role to tune molecular packing and miscibility with PC$_{71}$BM. Therefore, fluorinated end-capped acceptor were introduced in the molecular design, which lowered surface tension and their miscibility with PCBM, As a result, the lateral and vertical morphology of the active layer was optimized. **BTID-2F** formed a hierarchical morphology in the active layer,

which will inspire more investigations on the effects of π-bridges and end acceptors for high performance OSCs.

In summary, by combining traditional molecular design strategy with fine tuning surface tension and miscibility with $PC_{71}BM$ by fluorination, we designed and synthesized three novel molecules, **BTID-0F**, **BTID-1F** and **BTID-2F**, with incremental fluorine atoms. The three molecules exhibited excellent molecular properties, such as low HOMO levels, good crystallinity and high hole mobility, because of well-designed molecular structures, including new gradient-decreased electron density π-bridges and proper polarity of aromatic acceptors. Through device measurement and morphology characterization, we emphasized the importance of fluorination to hierarchical morphology, surface enrichment and directional vertical phase distribution. The optimal morphology was beneficial to charge transfer, charge collection and recombination suppression, which reduced the loss of $V_{oc}$, $J_{sc}$ and FF simultaneously. As a result, a record PCE of 11.3% was obtained in inverted OSCs based on small molecules, with $V_{oc}$ of 0.95 V, $J_{sc}$ of 15.7 mA cm$^{-2}$ and FF of 76%.

## Methods

**Solar cell fabrication and measurements.** Inverted devices were fabricated with a structure of glass/ITO/ZnO/donor:acceptor/MoO$_x$/Ag. The ZnO precursor solution was prepared by dissolving 0.14 g of zinc acetate dihydrate (Zn(CH$_3$COO)$_2$·2H$_2$O, 99.9%, Aldrich) and 0.5 g of ethanolamine (NH$_2$CH$_2$CH$_2$OH, 99.5%, Aldrich) in 5 ml of 2-methoxyethanol (CH$_3$OCH$_2$-CH$_2$OH, 99.8%, J&K Scientific). Patterned ITO glass with a sheet resistance of 15 Ω sq$^{-1}$ was purchased from CSG HOLDING Co., Ltd. The ITO-coated glass substrates were cleaned by ultrasonic treatment in detergent, DI water, acetone and isopropyl alcohol under ultrasonication for 20 min at each step. A thin layer of ZnO precursor was spin-coated at 5,000 r.p.m. onto the ITO surface. After being baked at 200 °C for 30 min, the substrates were transferred into a nitrogen-filled glove box. The mixture of small molecules and PC$_{71}$BM with total concentration ca. 18 mg ml$^{-1}$ stirred at 60 °C in chloroform for ca. 0.5 h until they are intensively dissolved. Subsequently, the active layer was spin-coated from chloroform solutions of blends. Finally, a layer of ca. 5 nm MoO$_x$ and then an Ag layer of ca. 160 nm was evaporated subsequently under high vacuum ($<1 \times 10^{-4}$ Pa).

Conventional devices were fabricated with a structure of glass/ITO/PEDOT:PSS/Ca/Al. The ITO-coated glass substrates were cleaned by the same procedure with inverted devices. A thin layer of PEDOT:PSS was spin-coated at 4,000 r.p.m. onto the ITO surface. After being baked at 150 °C for 15 min, the substrates were transferred into a nitrogen-filled glove box. The mixture of small molecules and PC$_{71}$BM with total concentration ca. 18 mg ml$^{-1}$ stirred at 60 °C in chloroform for ca. 0.5 h until they intensively dissolved. Subsequently, the active layer was spin-coated from blend chloroform solutions of small molecules and PC$_{71}$BM. Finally, a layer of ∼20 nm Ca and then 100 nm Al layer was evaporated under high vacuum ($<1 \times 10^{-4}$ Pa).

Device $J–V$ characteristics was measured under AM 1.5 G (100 mW cm$^{-2}$) using a Newport Thermal Oriel 91159A solar simulator. Light intensity is calibrated with a Newport Oriel PN 91150V Si-based solar cell. $J–V$ characteristics were recorded using a Keithley 2400 source-measure unit. Typical cells have device areas of approximately 4 mm$^2$. A mask with well-defined area was used to measure the $J–V$ characteristics as well. EQEs were performed in air with an Oriel Newport system (Model 66902) equipped with a standard Si diode. Monochromatic light was generated from a Newport 300 W lamp source. We have used mask for BTID-2F, the errors are in 5%.

Supplementary Methods including: characterization methods: (1) molecular structure characterization and calculation (nuclear magnetic resonance, mass spectrometry spectra, discrete Fourier transform); (2) molecular properties characterization (dielectric constant, UPS, ultraviolet–vis spectra, CV, UPS, contact angle); (3) TEM, AFM, XPS and in-depth XPS characterization; (4) GIWAXS characterization; (5) RSoXs characterization; and (6) $J_{ph}$ and mobility measurements. Calculation methods: (1) calculations of ionization potential of donor (IP$_D$) from ultraviolet photoelectron spectroscopy (UPS); (2) calculation of the coherence length ($L_c$) of PC$_{71}$BM and small molecules; (3) calculation of domain size and purity from RSoXs; (4) calculation of surface enrichment degrees; (5) calculation of surface D:A ratio; and (6) calculations of mobility measured from space-charge limited current. Synthesis methods including: materials and synthesis.

**Data availability.** All relevant data are available from the authors.

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

## Acknowledgements

We acknowledge the financial support by the Ministry of Science and Technology of China (No 2016YFA0200700), the National Natural Science Foundation of China (Grant Nos 21534003, 91427302, and 21504066) and the 'Strategic Priority Research Program' of the Chinese Academy of Sciences (Grant No XDA09040200). X-ray data was acquired at beamlines 7.3.3 and 11.0.1.2 at the Advanced Light Source, which is supported by the Director, Office of Science, Office of Basic Energy Sciences, of the U.S. Department of Energy under Contract No DE-AC02-05CH11231.

## Author contributions

D.D. designed, synthesized and characterized materials and fabricated devices based on BTID-2F. Y.Z. performed device fabrication of BTID-0F and BTID-1F. B.X. helped device fabrication. J.Z. performed GIWAXS data analysis. Z.W. and W.M. performed GIWAXS/RSoXs measurements and data analysis. L.Z. and Z.W. performed theoretical simulation. J.F. performed dielectric constant measurements. D.D., K.L. and Z.W. prepared manuscript. All authors discussed and commented on the paper. K.L. and Z.W. supervise the project.

## Additional information

**Competing financial interests:** The authors declare no competing financial interests.

