## [Peer Review File · Nature Communications]

Reviewer #1 (Remarks to the Author)

Review report for NCOMMS-16-14107-T

This work designed and prepared three F-containing small molecules, BTID-0F, BTID-1F and BTID-2F, and used them as the donor with PCBM for OPV devices, a high ~11% PCE was achieved. The results are definitely interesting and could be published in this journal, if the following issues are addressed:

1. First, there are many statements which could be more accurate or avoiding misunderstanding, such as "Enhanced surface enrichments and more unitary vertical phase gradation induced by fluoridation effectively blocks electrons and facilitates charge collection...", "heightening ionization potential of the donor...", "the contact angles of the three materials in chloroform on ZnO/ITO substrates. The increase of contact angle by fluoridation illustrated the reduced surface free energies.", "a novel morphology model was proposed", etc

2. It is stated that "Notably, the active layers of all devices were obtained without any additives and post-treatment" and the title of the Table 1 is "Optimized photovoltaic performance". Also no details of the device fabrication such as different conc of the material solution for the coating, different solvent, different temperatures etc are presented. Furthermore, these statement seems not consistent each other.

3. In the supporting information, in line 349 (table S2), the average PCE of BTID-2F is higher than its best results.

4. In the ms, line 33, the authors wrote that "the highest report is 7.88%" for small-molecules based inverted devices, while, a PCE of 8.84% for a small-molecules based inverted devices was reported in 2015. (DOI: 10.1021/acsami.5b05317).

5. In the Table 1, BTID-2F shows the highest Voc of 0.95 V, BTID-0F shows the lowest Voc of 0.91 V for their inverted devices. In the table S2, for their conventional devices, BTID-0F shows much higher Voc than that of BTID-2F. The authors could explain it more specific, as the readers will be very interested in it.

6. In line 159 (ms), the π - π stacking distance of BTID-2F is shorter than that for BTID-0F (also for their blend films in the table S4), and these should be beneficial for charge transport. While, in line 221, the hole mobility of BTID-2F is lower than that for BTID-0F. The authors are recommended to explain it.

7. In the supporting information, in line 209, the formula of compound 6 is C₉H₅FO₂ where the number of H atom is 5, while the sum of H atom is 6 from the HNMR, and the same question was found for the compounds 10, 12, and all the three target molecules BTID-0F/ BTID-1F/BTID-2F. In addition, the ¹³CNMR of some compounds are missing.

Reviewer #2 (Remarks to the Author)

The manuscript by Deng et al. focuses on the impact of fluorination of some conjugated donor molecules on the solar cell performances, announcing a 11.3% efficiency of the best performing devices. The authors have synthesized three molecular semiconductors with benzodithiophene core that differ by the presence of fluorinated end capping acceptor units. The number of fluorine atoms in the end capping units is shown to influence the electronic and structural properties of the materials in thin films. However, for the most fluorinated system (BTID-2F), the length of the solubilizing side chains was also increased, possibly to counterweight its insufficient solubility due to fluorination. The experimental work includes the synthesis of the semiconductors and their characterization using a large battery of methods (OPV device characterization using GIXD, AFM, TEM, X-PS, RsoXS,...).

Overall, this is a complete work which is certainly worth to be published. However, this referee is

not convinced that the novelty in the approach or the broad impact of the work are sufficient to warrant publication in Nat. Com. (a journal such as Chem. Mater. might be more adapted). Below are some comments to the authors.

1) Preliminary remark : Throughout the manuscript, the authors use the chemical term "fluoridation". This is incorrect as it refers to the F- species, not to atomic fluorine as used in the manuscript. The correct expression is "fluorination".

2) Influence of fluorination on morphology.

GIXD has been performed on the thin films. All films exhibit the same kind of patterns with a dominant in-plane pi-stacking peak and a sequence of three reflections in out of plane orientation. From such a pattern, the authors claim that the films have an edge-on orientation. However, such terminology is classical for polymers whereas for oligomers, it is not adapted. Indeed, the pi-stacking lying in-plane can be observed in both the cases where the molecule has its long axis standing or lying on the substrate. From the data, the authors do not explain if the low angle reflection in the range 0.4-0.45 Å⁻¹ corresponds to the long axis periodicity or to the short axis periodicity (along the solubilizing alkyl side chains). It is essential to clarify the orientation and the packing of the molecules on the substrate before drawing some conclusions on the impact of fluorination of the device performances.

As a remark, for polymers, so-called "edge-on orientation" is known to be rather unfavorable for OPV device performances and situations where the pi-stacking is normal to the substrate plane are usually sought for.

3) In page 4, the authors indicate that fluorination strengthen molecular interactions. This referee would rather say that usually fluorination reduces the solubility of the molecules in the solvents e.g. ODCB used for film fabrication, hence, promotes aggregation and eventually phase separation in presence of PCBM. This is not equivalent to say that fluorinated molecules have stronger interactions in the solid state.

4) The authors demonstrate that some vertical phase separation occurs in the blends of the molecules with PCBM and this phase separation is more pronounced for fluorinated molecules. Similar effects were demonstrated for polymers (S. J. Kim et al. Adv. Mater. 2010, 22, 1355). Ade and coworkers have also demonstrated that fluorination of semiconducting polymers yield high performances in devices, regardless of the morphology (J. R. Tumbleston et al., Adv. Funct. Mat. 2013, 23, 3463). Therefore, the debate on the exact origin of the influence of fluorination on device performances is somehow controversial. This referee is not convinced that this study does alleviate this controversy.

5) The technique of resonant soft X-ray scattering is not classical for the readership familiar with OPV. The whole explanation in page 7 is therefore hard to understand. This referee did not find the details on the experimental conditions for RSoXS in the supporting information.

Reviewer #3 (Remarks to the Author)

The authors designed and synthesized three small molecule donor materials for organic photovoltaic applications based on thiophene-substituted benzodithiophene core, with systematically incremental fluorine atoms incorporated in the end-capped acceptor units. Using inverted solar cell configuration, they have demonstrate the device power efficiency of 8.3%, 10.4% and 11.3% for these three type of donors with PC71BM as acceptor. Overall, it is a great original work in this field. The reviewer recommend publication of this work after the authors address suggested minor revisions (see the last part of the review comments).

In this manuscript, fundamental properties relevant to device physics were characterized, including ionization potential, energy levels, absorption spectra. Device performance was investigated. Active layer film morphology was studied with grazing-incidence wide angle x-ray scattering (GIWAXS), atomic force microscopy (AFM), transmission electron microscopy (TEM) and resonant soft x-ray scattering (RSoXS), in order to obtain semicrystalline information and donor acceptor phase separation size and domain purity. Film morphology in surface normal direction is probed

with XPS depth profiling. It shows overall good data quality with reasonable data statistics and the morphology study is used to explain the observed device performance difference for three types of donors.

Questions and suggested revisions:

1. In Fig. 4, are the AFM images phase images or height images? If it's phase images, what does the contrast stands for considering this surface is supposed to be pure donor according to XPS profiling; if it's height images, the height scale bar is required to help reader to justify the roughness.
2. In the GIWAXS results (Fig. 3), the intensity color bar for intensity scaling should be given and the data normalization to film thickness and sample size should be described in the text or caption.
3. The authors need to describe what energy of soft x-rays were used for RSoXS experiments and explain why that energy is chosen. It is needed to consider that the surface roughness is high for the active layers, and its lateral distribution could cause scattering in RSoXS data, which might not be from phase separation of donor and acceptor. Therefore, in order to claim the hierachical domain size between donor and acceptor phase separation and further to analyze domain purity, it is necessary to demonstrate that the scattering contrast seen in Fig. 4g is indeed due to phase separation, at minimum by also including the scattering intensity at off resonant energy (for example 270 eV).

Response to the comments of reviewer

We would like to thank the referees for spending time on this paper and providing invaluable comments which substantially helped improving the quality of the paper. The manuscript has been carefully revised according to the comments.

Reviewer 1

General comments: This work designed and prepared three F-containing small molecules, BTID-0F, BTID-1F and BTID-2F, and used them as the donor with PCBM for OPV devices, a high ~11% PCE was achieved. The results are definitely interesting and could be published in this journal.

Comment 1: First, there are many statements which could be more accurate or avoiding misunderstanding, such as "Enhanced surface enrichments and more unitary vertical phase gradation induced by fluoridation effectively blocks electrons and facilitates charge collection...", "heightening ionization potential of the donor...", "the contact angles of the three materials in chloroform on ZnO/ITO substrates The increase of contact angle by fluoridation illustrated the reduced surface free energies..", "a novel morphology model was proposed", etc

Reply: Thanks for the reviewer's comments. The statements of the whole text have been carefully checked, and the sentences easy to cause misunderstanding are corrected in the revision.

Comment 2: It is stated that "Notably, the active layers of all devices were obtained without any additives and post-treatment" and the title of the Table 1 is "Optimized photovoltaic performance".

Also no details of the device fabrication such as different conc of the material solution for the coating,

different solvent, different temperatures etc are presented. Furthermore, these statement seems not consistent each other.

Our reply and revision: Thanks for the reviewer’s comments. We have tried different conditions, and found the optimized condition is that without any additives and post-treatment. For instance, the conditions of adding various additives, thermal annealing and solvent annealing have been tested for inverted device based on BTID-2F, but the efficiencies are all lower than that without any post-treatment, and details are shown in **Table R1**. The detailed data for optimization of donor to acceptor (D: A) ratios are shown in **Table. R2**, and added in Supplementary Information as Table S2 in the revision. Furthermore, the details of the inverted and conventional device fabrication are added in solar cell fabrication and measurements section in the revision as suggested.

Table R1 Optimization of additives, concentrations (Conc), thermal annealing (TA) and solvent annealing (SVA) for BTID-2F (D: A=1.3:1)

treatment	Details	Conc mg/ml	Voc (V)	Jsc (mA/cm ²)	FF(%)	PCE (%)
None	None	17	0.95	14.8	76.4	10.8
None	None	18.5	0.95	15.7	76.0	11.3
Additives	0.25%DIO	17	0.94	15.5	71.3	10.4
	0.25%CN	17	0.97	13.2	69.6	8.9
	CF	17	0.92	14.8	71.7	9.74
SVA	Pyridine	17	0.93	15.0	68.0	9.51
1 min	CS ₂	17	0.93	14.6	70.4	9.54
	Toluene	17	0.94	15.2	71.4	9.51

TA (10 minutes)	60 °C	17	0.96	14.6	73.3	10.3
	80 °C	17	0.94	14.7	71.5	9.9

Table R2 Optimization of D: A ratios for BTID-2F, BTID-1F, BTID-0F

Donors	D:A	Voc (V)	Jsc (mA/cm ²)	FF (%)	PCE (%)
BTID-2F	1.1:1	0.94	15.8	74.0	11.0
	1.3:1	0.95	15.7	76.0	11.3
	1.5:1	0.95	15.3	75.0	10.9
BTID-1F	1.3:1	0.94	15.5	68.5	10.0
	1.5:1	0.94	15.3	72.0	10.4
	1.7:1	0.93	15.1	65.6	9.2
BTID-0F	1.3:1	0.93	14.6	51.2	6.9
	1.5:1	0.93	14.0	64.0	8.3
	1.7:1	0.92	14.6	60.8	8.1

Comment 3: In the supporting information, in line 349 (table S2), the average PCE of BTID-2F is higher than its best results.

Reply: Thanks for the reviewer's comments. We have corrected this mistake in the revision (Table S2, Page 18 in SI).

Comment 4: In the ms, line 33, the authors wrote that "the highest report is 7.88%" for small-molecules based inverted devices, while, a PCE of 8.84% for a small-molecules based inverted

devices was reported in 2015. (DOI: 10.1021/acsami.5b05317).

Reply: Thanks for the reviewer's comments. We have updated the data in page 2 in the revision.

Comment 5: In the Table 1, BTID-2F shows the highest V_{oc} of 0.95 V, BTID-0F shows the lowest V_{oc} of 0.91 V for their inverted devices. In the table S2, for their conventional devices, BTID-0F shows much higher V_{oc} than that of BTID-2F. The authors could explain it more specific, as the readers will be very interested in it.

Reply: The difference of V_{oc} is an important phenomena that need to be further investigated for difference device structures. We are also quite interested in the V_{oc} loss and its relation to the other factors besides energy levels. In this manuscript, we should emphasize the disparity of V_{oc} based on the three molecules is very small (all in the range of 0.91-0.96V). Hence, the tendency could be easily changed by the different device structure.

We have done some preliminary work on the reason of V_{oc} differences. We carried out capacitance–voltage measurement and analyzed it by Mott–Schottky (MS) analysis reported in the literature (*Adv. Energy Mater.* 2012, **2**, 82):

$$c^{-2} = \left(\frac{2}{q\epsilon N}\right)(V_{bi} - V)$$

Where V_{bi} is build-in potential, N impurity concentration. The C^{-2} - V plots of conventional (Conv) and inverted (Inv) devices based on the three molecules are shown in **Fig. R1**. From the detailed data shown in **Table. R3** ($\Delta V_{oc}=(V_{oc})_{Inv}-(V_{oc})_{Conv}$, $\Delta V_{bi}=(V_{bi})_{Inv}-(V_{bi})_{Conv}$), it could be easily found: for BTID-0F, the V_{bi} is larger for conventional devices, while for BTID-2F, the V_{bi} is larger for inverted devices; what's more, the change tendency of V_{oc} is well consistent with the change tendency of V_{bi} . Consequently, the changed V_{oc} should be resulted from the changed V_{bi} .

Table. R3 Detailed results of Capacitance-Voltage measurements

Donor	Devices	V_{oc}	V_{bi}	N	ΔV_{oc}	ΔV_{bi}
BTID-0F	Conv	0.955	0.746	1.55E22	-0.029	-0.035
	Inv	0.926	0.711	2.20E22		
BTID-1F	Conv	0.936	0.738	1.68E22	0.004	0.002
	Inv	0.940	0.740	2.24E22		
BTID-2F	Conv	0.905	0.700	9.47E21	0.049	0.083
	Inv	0.954	0.783	1.58E22		

**Fig. R1** C^{-2} and voltage plots of conventional and inverted devices

Some literatures reported the changed V_{bi} could be resulted from the change of the Fermi level of the active layer (ΔE_F^p) because of recombination (*Phys. Status Solidi RRL* 5, 2011, 7, 247; *Org Electron*, 2013, 14, 3083). To further analyze the change of V_{bi} , we studied ΔE_F^p for the inverted and conventional devices.

$$\Delta E_F^p = k_b T \ln \frac{N_{inv}}{N_{conv}}$$

However, the calculated $\Delta E_F^p < 20$ mV, which means besides the change in Fermi level of the active layer, there are other factors to influence V_{bi} , for example: different electrode contact and work function modification resulted from the different surface enrichment, and dipole change resulted from the vertical phase separation.

Based on the above mentioned experiments, one sentence is added in the revision: “Moreover, the vertical distribution and surface enrichment reduced recombination and increased the build-in potential of inverted devices in comparison of conventional devices, which thereby further reduced V_{oc} loss.” (Page 8)

Comment 6: In line 159 (ms), the π - π stacking distance of BTID-2F is shorter than that for BTID-0F (also for their blend films in the Table S5), and these should be beneficial for charge transport. While, in line 221, the hole mobility of BTID-2F is lower than that for BTID-0F. The authors are recommended to explain it.

Reply: Thanks for the reviewer’s comments. The average ability is 8.7×10^{-4} , 6.4×10^{-4} and $3 \times 10^{-4} \text{ cm}^2 \text{ V}^{-1} \text{ s}^{-1}$ for BTID-0F, BTID-1F and BTID-2F respectively, and they are in the same order of magnitude, and could be influenced by other factors. Besides π - π stacking distance, there are many other factors to influence the hole motilities, such as charge transfer pathway, crystallinity and phase continuity. Therefore, the smaller π - π stacking distance is not always equals to the higher hole mobility, which also reported in the other literatures (Adv. Mater. 2012, 24, 6457; J. Am. Chem. Soc. 2014, 136, 2135).

In our case, the molecular orientation might be the main reason for a little bit lower mobility of BTID-2F. As shown in **Fig. 3a-c**, the main packing mode of the three molecules is edge-on, and however, they also exhibit some face-on packing mode. We have calculated the face-on to edge on ratios according to the literature (*Nature Photonics*, 2015, 9, 403). With fluorination, the molecules are prone to more edge-on in comparison of face-on (**Fig. R2**), and probably, this could be the main reason for the decreased mobility with fluorination. In addition, we also mentioned the impact of the

length of alkyl chains on the hole mobility, since BTID-2F have longer alkyl chains than that of BTID-0F.

Fig. R2 The edge on and face on packing mode ratios of the pristine films

The explanations have been added in the revision (page 8) as: “The hole mobility of pristine film for BTID-0F, BTID-1F and BTID-2F were 8.7×10^{-4} , 6.4×10^{-4} and $3 \times 10^{-4} \text{ cm}^2 \text{ V}^{-1} \text{ s}^{-1}$, the decrease of hole mobility with fluorination should be mainly resulted from their decreased face-on packing ratios (Supplementary Table. S5) and longer alkyl chain lengths.”

Comment 7: In the supporting information, in line 209, the formula of compound 6 is $\text{C}_9\text{H}_5\text{FO}_2$ where the number of H atom is 5, while the sum of H atom is 6 from the HNMR, and the same question was found for the compounds 10, 12, and all the three target molecules BTID-0F/BTID-1F/BTID-2F. In addition, the ^{13}C NMR of some compounds are missing.

Reply: Thanks for the reviewer’s comments. We have carefully checked the chemical shifts and H atoms, and made the correction. The lack of ^{13}C NMR spectrums have been added in synthesis section of supplementary Information.

Reviewer 2:

General comments: The manuscript by Deng et al. focuses on the impact of fluorination of some conjugated donor molecules on the solar cell performances, announcing a 11.3% efficiency of the best performing devices. The authors have synthesized three molecular semiconductors with benzodithiophene core that differ by the presence of fluorinated end capping acceptor units. The number of fluorine atoms in the end capping units is shown to influence the electronic and structural properties of the materials in thin films. However, for the most fluorinated system (BTID-2F), the length of the solubilizing side chains was also increased, possibly to counterweight its insufficient solubility due to fluorination. The experimental work includes the synthesis of the semiconductors and their characterization using a large battery of methods (OPV device characterization using GIXD, AFM, TEM, X-PS, RsoXS,...).

Overall, this is a complete work which is certainly worth to be published. However, this referee is not convinced that the novelty in the approach or the broad impact of the work are sufficient to warrant publication in Nat. Com. (a journal such as Chem. Mater. might be more adapted)

Reply: We emphasize the novelties and importance of this manuscript in Discussion part in the revision. Besides of those, although researchers have already synthesized thousands of small molecules for organic solar cells, their efficiency are not high enough comparing with that of polymers, but the reason behind that are not well understood yet. This manuscript reported two novel aspects for the molecular design, which will be important for guiding the design of high performance molecules:

(1) Design a novel π -bridge, which provide an inner gradient-decreased electron density distribution molecular design strategy. The new π bridge together with TBDT units and end-capped acceptor

presents an inner gradient-decreased electron density distribution. This strategy benefits for backbone hole transfer, which hasn't been reported before in the molecular design for organic solar cells, as far as we know.

(2) Design two novel fluorinated end-capped acceptors, and found fluorination of the end-capped acceptor is very efficient to optimize the lateral and vertical morphology of active layers. As a result, a record performance of 11.3% is obtained, with a V_{oc} of 0.95 V, J_{sc} of 15.7 mA cm⁻² and FF of 76% in inverted solar cells for BTID-2F.

In addition, as the reviewer refers to, fluorination would lower the molecular solubility, and the increased alkyl chains length attached to TBDT is to increase the molecular solubility, which we have mentioned in page 3.

Comments 1: Preliminary remark: Throughout the manuscript, the authors use the chemical term "fluoridation". This is incorrect as it refers to the F- species, not to atomic fluorine as used in the manuscript. The correct expression is "fluorination".

Reply: Sincerely thanks for the reviewer's comments. We have checked the manuscript carefully and made the corrections.

Comments 2: Influence of fluorination on morphology.

GIXD has been performed on the thin films. All films exhibit the same kind of patterns with a dominant in-plane pi-stacking peak and a sequence of three reflections in out of plane orientation. From such a pattern, the authors claim that the films have an edge-on orientation. However, such terminology is classical for polymers whereas for oligomers, it is not adapted. Indeed, the pi-stacking

lying in-plane can be observed in both the cases where the molecule has its long axis standing or lying on the substrate. From the data, the authors do not explain if the low angle reflection in the range 0.4-0.45 Å⁻¹ corresponds to the long axis periodicity or to the short axis periodicity (along the solubilizing alkyl side chains). It is essential to clarify the orientation and the packing of the molecules on the substrate before drawing some conclusions on the impact of fluorination of the device performances.

As a remark, for polymers, so-called "edge-on orientation" is known to be rather unfavorable for OPV device performances and situations where the pi-stacking is normal to the substrate plane are usually sought for.

Our reply: Thanks for the reviewer's comments.

As the reviewer mentioned, besides face on and edge on packing modes, for oligomers, there is another packing mode named end-on, whose backbone (long axis) is perpendicular to the substrate (*J. Am. Chem. Soc.* 2013, 135, 9644). The packing mode and GIXRD peaks of end-on and edge-on as shown in **Fig. R3**, using thiophene oligomers as an example. As in this mode, both the (100) (200) peaks and (010) peak should be lying in the in plane direction. As in our manuscript, an obvious strong (010) peak is lying in the in plane direction, while the strong (100) (200) (300) peaks are lying in the out of plane direction. Hence, the three molecules in our manuscript mainly adopt edge-on packing.

Fig. R3 The packing and GIXRD peaks schematic of thiophene oligomers.

The so-called “face-on” orientation is indeed desired for OPV applications, for it probably shortens the charge transport path. However, there are still various literatures reported high efficiency with edge-on molecular packing mode (*J. Am. Chem. Soc.*, **2015**, 137, 3886). For the electric coupling, domain size and domain purity also play important role in the device performance. Furthermore, we have not found a good method to switch the orientation. We expect, if the “face-on” orientation can be achieved and the original morphology could be kept in certain range, the efficiency of the devices can be further improved.

In this manuscript, we focused on the discussion of GIWAXS of blend films and did not explain 0.4-0.45 \AA^{-1} in the pristine films of BTID-2F for the following three reasons: (1) The peak in the range 0.4-0.45 \AA^{-1} is appeared only for BTID-2F pristine film but disappeared in its blends with PC₇₁BM. However, the device performance mainly depends on its morphology of blends. (2) The peak of 0.4-0.45 \AA^{-1} (the corresponding d-spacing is 13.5 \AA) could not be correspond to (001) peak,

and it is possibly corresponds to some certain crystal plane. For the d-spacing of (100) is *ca.* 19 Å for BTID-2F (corresponds to the short axis periodicity, which is along the solubilizing alkyl side chains), while the d-spacing of (001) (corresponds to the long axis periodicity) should be longer than 19 Å. (3) We have tried but failed to grow single crystals for BTID-2F. Hence, we couldn't make sure the definitely crystal plane of the peak of 0.4-0.45 Å⁻¹.

With the comments of the reviewer, we have modified the sentence in Hierarchical morphology section: "The differences in d-spacing in the (100) direction were ascribed to the varying lengths of the alkyl (corresponds to the short axis periodicity)."

Comments 3: In page 4, the authors indicate that fluorination strengthen molecular interactions. This referee would rather say that usually fluorination reduces the solubility of the molecules in the solvents e.g. ODCB used for film fabrication, hence, promotes aggregation and eventually phase separation in presence of PCBM. This is not equivalent to say that fluorinated molecules have stronger interactions in the solid state.

Reply: Thanks for the reviewer's comments. We have revised "indicate that fluorination strengthen molecular interactions" into "fluorination led to dense and ordered π - π stacking" in page 4 in the revision.

Comments 4: The authors demonstrate that some vertical phase separation occurs in the blends of the molecules with PCBM and this phase separation is more pronounced for fluorinated molecules. Similar effects were demonstrated for polymers (S. J. Kim et al. Adv. Mater. 2010, 22, 1355). Ade and coworkers have also demonstrated that fluorination of semiconducting polymers yield high

performances in devices, regardless of the morphology (J. R. Tumbleston et al., *Adv. Funct. Mat.* 2013, 23, 3463). Therefore, the debate on the exact origin of the influence of fluorination on device performances is somehow controversial. This referee is not convinced that this study does alleviate this controversy.

Reply: Thanks for the reviewer's comments and the recommendation of two excellent literatures.

As reviewer refers to, in comparison of non-fluorinated molecules, fluorinated molecules exhibit pronounced phase separation (*Adv. Mater.* 2010, 22, 1355). However, for polymer PBnDT-FTAZ, the device performance exhibits insensitivity to the morphology in a certain range, as illustrated in Ade and You Wei's work (*Adv. Funct. Mat.* 2013, 23, 3463). By reading the two literatures carefully, we found there is no conflict between the two literatures, and both of the literatures emphasize the importance of morphologies. The reasons are summarized as followings:

(1) In comparison of non-fluorinated molecules, most fluorinated molecules exhibited better molecular packing, larger domain size and higher domain purity. The authors were inclined to ascribe the device performances to the improved morphologies (review, *Progress in Polymer Science*, **2015**, 47, 70; review, *Polymers*, **2016**, 8, 11). This statement was also illustrated in Wei You and Ade's work (*J. Am. Chem. Soc.*, **2013**, 135, 1806).

(2) For fluorinated polymer PBnDT-FTAZ, Ade and Wei You investigated the solvent effects on morphology and device performance. They emphasized the insensitivity was "in a certain range" (*Adv. Funct. Mat.* 2013, 23, 3463). Beyond the "certain range", the device performance decreased dramatically. For the device performance of PBnDT-FTAZ processed with CB (3.1%) is much lower than those processed with DCB (6.4%) and frozen TCB (7.0%). They attributed this phenomenon to its much lower domain purity (<0.4, while the others *ca.* 0.8-1). In addition, there are many

literatures reported fluorinated polymers and small molecule were sensitive to morphologies resulted from substrate temperature (*Nature Commun.* DOI: 10.1038/ncomms6293), DIO concentration and thermal annealing with slight disturbance (*Adv. Funct. Mater.* 2013, 23, 5019; *Adv. Energy Mater.* 2015, 5, 1500877). Furthermore, for the non-fluorinated small molecule based on DPP, it exhibited insensitivity to morphology whether processed with CF (*Adv. Energy Mater.* 2013, 3, 724) or CB (*ACS Appl. Mater. Interfaces*, 2013, 5, 2033). Hence, no matter for the fluorinated or non-fluorinated molecules, the device performance would be kept “in a certain range,” and whether the device performance is insensitive to morphology should depend on materials system.

However, in this manuscript, we mainly focused on the comparison between the fluorinated-materials and non-fluorinated materials. In fact, for our materials system, hierarchical morphology, surface enrichment and vertical phase separation are quite important for the inverted device performance based on BTID-2F. The analysis are as below:

(a) Evidence of the role of hierarchical morphology. When the substrate temperature is increased from 28 °C (cool) to 40 °C (hot), its hierarchical morphology disappeared, as shown in the TEM images and RSoXs images (**Fig. R4**), and the main domain size is decreased from 53 nm to 17 nm. Though the smaller domain size facilitate charge separation, however, the device performance based on the cool substrate (the hierarchical morphology) is much higher than that on the hot substrate (**Table. R4**). Hence, the hierarchical morphology would reduce the recombination loss in our materials system.

Table R4 The device performance is sensitive for inverted device based on BTID-2F

	V_{oc} (V)	J_{sc} (mA/cm ²)	FF (%)	PCE(%)
Hot substrate	0.95	15.1	71	10.2

Fig. R4 TEM and RSoXs images of cool and hot substrate based on BTID-2F

(b) Evidence of the role of surface enrichment and vertical phase separation. In the previous manuscript, we have emphasized the different device performance between the inverted (Inv) and conventional (Conv) devices. After fluorination, the lateral morphologies (molecular packing, TEM, domain size, domain purity) changed in the same tendency whether on ZnO or PEDOT: PSS substrate (Supplementary Conventional photovoltaic device performance and related characterization section); while in the vertical direction, electron-blocking layer formed on the active layer and the hole transported path from the active layer to surface is only favored by inverted devices. This could explain the phenomenon that BTID-2F obtained a higher inverted device performance while BTID-0F obtained a higher conventional device performance. Hence, the surface enrichment and vertical phase separation play important role to enhance the inverted device performance based on

BTID-2F.

Table R4 The inverted device performance is sensitive to surface enrichment and vertical phase separation

Donor	Devices	V_{oc} (V)	J_{sc} (mA/cm ²)	FF (%)	PCE(%)
BTID-0F	Conv	0.96	13.7	73.0	9.5
	Inv	0.93	14.0	64.0	8.3
BTID-1F	Conv	0.94	13.9	72.0	9.4
	Inv	0.94	15.3	72.0	10.4
BTID-2F	Conv	0.91	14.1	73.0	9.3
	Inv	0.95	15.7	76.0	11.3

Hence, in our opinion, in comparison of non-fluorinated materials with fluorinated materials, fluorination plays an important role in the morphology. The sensitivity or insensitivity to morphology (resulted from device fabrications, such as additives and solvents) should be related to materials system.

For a better understanding of the influence of fluorination on the morphology, we add the “Evidence of the role of hierarchical morphology” and modified the “Evidence of the role of surface enrichment and vertical phase separation” in the Supporting information.

Comments 5: The technique of resonant soft X-ray scattering is not classical for the readership familiar with OPV. The whole explanation in page 7 is therefore hard to understand. This referee did

not find the details on the experimental conditions for RSoXS in the supporting information.

Reply: Thanks for the reviewer's comments. There was only a small paragraph in the supplementary characterization section to introduce RSoXS. To make it more comprehensive, we have added the following words to show how the relative purity is extracted from **RSoXS** in the Supplementary Calculation of domain size and purity from RSoXS section.

The details: The scattering profiles can be processed further to extract the domain purity of an assumed two phase system through the total scattering intensity (TSI):

$$\text{TSI} = \int_0^{\infty} I(q)q^2 dq = 2\pi^2 \Delta\rho_{12}^2 v_1 v_2 V,$$

where $\Delta\rho_{12}$ is the difference in electron density between the two phases, v_i is the material volume fraction of each domain and V is the illuminated volume. Domain contrast affect the TSI and thus can be used as a measure of domain purity. In resonant soft x-rays region, ρ can be extended as

$$\rho = \frac{f_1 - if_2}{V} = \sum_j n_j (f_{1j} + f_{2j}) = \alpha E^2 (\delta - i\beta),$$

where $f_1 - if_2$ are the complex elemental scattering factors, n_j are the number densities for the j^{th} element in each component, and α is a constant factor. Thus, $\Delta\rho_{12}^2 = \alpha^2 E^4 \Delta n_{12}^2$. $\Delta\rho$ indicates the contrast between domains (not materials). If the domains are purer (materials are less mixed within domains), the higher contrast will lead to higher TSI. As Δn scales with purity, relative domain purity can be readily obtained from TSI.

Reviewer 3:

General comments: The authors designed and synthesized three small molecule donor materials for organic photovoltaic applications based on thiophene-substituted benzodithiophene core, with systematically incremental fluorine atoms incorporated in the end-capped acceptor units. Using

inverted solar cell configuration, they have demonstrate the device power efficiency of 8.3%, 10.4% and 11.3% for these three type of donors with PC₇₁BM as acceptor. Overall, it is a great original work in this field. The reviewer recommend publication of this work after the authors address suggested minor revisions (see the last part of the review comments).

In this manuscript, fundamental properties relevant to device physics were characterized, including ionization potential, energy levels, absorption spectra. Device performance was investigated. Active layer film morphology was studied with grazing-incidence wide angel x-ray scattering (GIWAXS), atomic force microscopy (AFM), transmission electron microscopy (TEM) and resonant soft x-ray scattering (RSoXS), in order to obtain semicrystalline information and donor acceptor phase separation size and domain purity. Film morphology in surface normal direction is probed with XPS depth profiling. It shows overall good data quality with reasonable data statistics and the morphology study is used to explain the observed device performance difference for three types of donors.

Comment 1: In Fig. 4, are the AFM images phase images or height images? If it's phase images, what does the contrast stands for considering this surface is supposed to be pure donor according to XPS profiling; if it's height images, the height scale bar is required to help reader to justify the roughness.

Reply: Thanks for the reviewer's comments. The AFM images we presented in the manuscript are phase images, and we have added the color bar in the Fig. 4 and supplementary Fig. S17.

The contrast of the phase images could be influenced by the different viscosity of different parts of one material or the different viscosity of different materials, as well as the hard parts and the soft parts of one materials or/and different materials. In order to give more intuitive images, we measured

the AFM of the pristine donors (**Fig. R5**). No matter for the pristine films or the blends, all the phase images exhibit the contrast. In our manuscript, the phase images are used to roughly estimate the size of crystalline parts because of their similar molecular backbone.

Fig. R5 AFM phase images of pristine and blend films

Comment 2. In the GIWAXS results (Fig. 3), the intensity color bar for intensity scaling should be given and the data normalization to film thickness and sample size should be described in the text or caption.

Reply: Thanks for the reviewer's comments. The intensity scaling is added in the revised manuscript. The intensity is normalized by the film thickness and samples size, which is added in the caption of the Fig. 3 and supplementary Fig. S14.

Comment 3. The authors need to describe what energy of soft x-rays were used for RSoXS experiments and explain why that energy is chosen. It is needed to consider that the surface

roughness is high for the active layers, and its lateral distribution could cause scattering in RSoXS data, which might not be from phase separation of donor and acceptor. Therefore, in order to claim the hierarchical domain size between donor and acceptor phase separation and further to analyze domain purity, it is necessary to demonstrate that the scattering contrast seen in Fig. 4g is indeed due to phase separation, at minimum by also including the scattering intensity at off resonant energy (for example 270 eV).

Reply: Thanks for the reviewer's comments. The photo energy of 284.2 eV is used for RSoXS measurement. As suggested by the reviewer, we have added the data of 270 eV in the revised manuscript (Supplementary Fig. S3) and shown in **Fig. R6**). The scattering peak at $q < 0.01 \text{ nm}^{-1}$ is observed for 270 eV (surface roughness). However, this peak does not appear at 284.2 eV. The significantly enhanced new peaks at $\sim 0.063, \sim 0.25 \text{ nm}^{-1}$ demonstrates multiple length scale phase separation in the blends.

Fig. R6 Comparison of TSI based on different photon energy choose for RSoXs

We have added a sentence to make it clear in Supplementary Calculation of domain size and purity from RSoXs section: “A photon energy of 284.2 eV was selected to provide high contrast and avoid high absorption which can lead to beam damage and fluorescence background.”

Reviewer #1 (Remarks to the Author)

The revision and replies have addressed my comments and it is fine with me for acceptance.

Reviewer #2 (Remarks to the Author)

The manuscript has been revised and might be published with minor corrections given below.

Line &176-177 : sentence to be rephrased, not clear.

... the domains increased in size...

Line 209 : ... enriched at the film surface.

Line 178 : the authors talk about fibrillar structures. However, this reviewer would rather talk about whiskers. Fibrils are highly elongated domains that cannot be observed in the AFM or TEM images.

Line 248 : fluoridation -> fluorination

Line 261 : and facilitated to -> and facilitated the

Line 442 :performed dielectric constant measurements.

Reviewer #3 (Remarks to the Author)

The comments of this referee have been addressed in this revised version. It is recommended that this manuscript be published on Nature Communications.

Response to the comments of reviewer (Second Revision)

Reviewer 1 (Remarks to the Author):

The revision and replies have addressed my comments and it is fine with me for acceptance.

Reviewer 2 (Remarks to the Author):

The manuscript has been revised and might be published with minor corrections given below.

Line &176-177 : sentence to be rephrased, not clear.

... the domains increased in size...

Line 209 : ... enriched at the film surface.

Line 178 : the authors talk about fibrillar structures. However, this reveiwer would rather talk about whiskers. Fibrils are highly elongated domains that cannot be observed in the AFM or TEM images.

Line 248 : fluoridation -> fluorination

Line 261 : and facilitated to -> and facilitated the

Line 442 :performed dielectric constant measurements.

Reply: We have revised the paper according to the reviewer's suggestion.

Reviewer 3 (Remarks to the Author):

The comments of this referee have been addressed in this revised version. It is recommended that this manuscript be published on Nature Communications.